# Two-Dimensional Soil Geometric Tortuosity Model Based on Porosity and Particle Arrangement

**Jin Gan** [1,2,3], **Zhiquan Yang** [1,2,3,*], **Zhiwei Zhang** [2,3,4], **Chaoyue Li** [1,2,3], **Yi Yang** [1,2,3], **Yingyan Zhu** [1,2,3,5], **Yanhui Guo** [1,2,3], **Renchao Wang** [6], **Bihua Zhang** [7], **Yingchao Fang** [8], **Dongliang Yu** [8], **Jie Zhang** [9], **Hao Liu** [10] **and Jiankun Su** [10]

1 Faculty of Public Safety and Emergency Management, Kunming University of Science and Technology, Kunming 650093, China; ganjin0619@163.com (J.G.); lichaoyue20@163.com (C.L.); kggtyy@163.com (Y.Y.); zh_y_y@imde.ac.cn (Y.Z.); guoyanhui0818@kust.edu.cn (Y.G.)
2 Key Laboratory of Geological Disaster Risk Prevention and Control and Emergency Disaster Reduction of Ministry of Emergency Management of China, Kunming 650093, China; 20192201095@stu.kust.edu.cn
3 Key Laboratory of Early Rapid Identification, Prevention and Control of Geological Diseases in Traffic Corridor of High Intensity Earthquake Mountainous Area of Yunnan Province, Kunming 650093, China
4 Faculty of Land Resources Engineering, Kunming University of Science and Technology, Kunming 650093, China
5 Institute of Mountain Hazards and Environment, Chinese Academy of Sciences, Chengdu 610041, China
6 School of Computer Science and Engineering, University of Electronic Science and Technology of China, Chengdu 611731, China; SupermanWang@uestc.edu.cn
7 Beijing Fibote Photoelectric Technology Co., Ltd., Beijing 100083, China; zhangbihua2008@126.com
8 Pipe China South West Pipeline Company, Chengdu 610041, China; fangyc@pipechina.com.cn (Y.F.); yudl@pipechina.com.cn (D.Y.)
9 Yunnan Institute of Geological Environment Monitoring, Kunming 650216, China; ynsghszj@163.com
10 Yunnan Aerospace Engineering Geophysical Detecting Co., Ltd., Kunming 650217, China; liuh@aerospace.net.cn (H.L.); sujk@aerospace.net.cn (J.S.)
* Correspondence: yzq1983816@kust.edu.cn

**Abstract:** Porosity and particle arrangement are important parameters affecting soil tortuosity, so it is of great significance to determine the intrinsic relationship between them when studying soil permeability characteristics. Theoretical derivation and geometric analysis methods are used to derive a two-dimensional geometric tortuosity model. The model is a function of particle arrangement parameters ($m$ and $\theta$) and porosity. An analysis of the model and its parameters shows that: (1) The arrangement of particles is one of the reasons for the different functional relationship between tortuosity and porosity, which proved that the tortuosity is not only related to the porosity but also affected by the particle arrangement. (2) The greater the anisotropy parameter $m$ is, the greater the tortuosity is, indicating $m$ varies when fluid passes through the soil from different sides resulting in different values of permeability. (3) The tortuosity increases with the increase in the blocking parameters $\theta$. (4) With increasing porosity, the influence of the parameters $m$ and $\theta$ on the tortuosity gradually decreases, suggesting that the influence of particle arrangement on tortuosity gradually decreases. The results presented here increase the understanding of the physical mechanisms controlling tortuosity and, hence, the process of fluid seepage through soil.

**Keywords:** particle arrangement; porosity; tortuosity; geometric analysis; two-dimensional; seepage

## 1. Introduction

Porous media materials exist in a wide range of fields, such as rocks and soils in mines [1,2] and in slopes [3]. As a typical porous medium material, soil has good permeability. Although there are many factors affecting the permeability of soil, such as porosity, particle arrangement, shape and size and fluid properties, it is generally believed that the distribution of pores in soil is the main factor that determines the permeability of soil [4].

The intercommunication of soil pores forms the channels for fluid seepage in the soil, but these channels are usually tortuous, and the degree of twists and turns is described by tortuosity.

Tortuosity is a term used to describe the sinuosity and interconnectedness of the pore space as it affects transport processes through porous media [5,6]. In recent years, the influence of tortuosity on the penetration process of fluids in soils has been increasingly studied. For example, the tortuosity effect was considered in penetration of grouting [7,8] and oil recovery for petroleum applications [9–11]. Therefore, tortuosity is an important factor to consider when attempting to understand the permeability of soil.

A model of tortuosity can be determined by experiment, numerical analysis, or geometric analysis [12–14]. Using the experimental method, Comiti [15] and Wyllie [16] derived an empirical model of tortuosity, using a fluid flow through fixed beds that are packed with particles of different shapes, which employed fitting parameters related to particle shape, and Wyllie suggested the parameter was between 2 and 3. Mota et al. [17] carried out experiments with a porous media composed of spherical particles, and their statistics suggested that the tortuosity increases as a power function with the decrease in porosity, but the model was only useful for transport phenomena analysis in granular beds.

Considering numerical analysis, Koponen et al. [18,19] obtained tortuosity models with and without considering the effective porosity of two-dimensional porous media by numerical simulation. For a given two-dimensional porous medium, the numerical analysis results showed that the tortuosity almost does not change when the lattice resolutions is different. Mayken et al. [20] used the lattice Boltzmann method (LBM) to generate two-dimensional porous media with different particle shapes and obtained corresponding expressions for tortuosity. Based on the analysis of these expressions, it was found that they have the same trend, but the numerical values were quite different.

Examples of using geometric analysis include Yu and Li [21] who first used the fixed arrangement of square particles to analyze and obtain a model of two-dimensional tortuosity. Plessis and Masliyah [22] used the concept of average volume to analyze and obtain a tortuosity model for isotropic porous media. Recently, a tortuosity model for different particle arrangements was obtained by Yan Han et al. [23].

The above tortuosity models are listed in Table 1. It can be seen from the table that, compared with the experimental method and numerical analysis, the tortuosity models derived using geometric analysis do not contain fitting parameters and, hence, can better reflect the physical mechanism of tortuosity. At the same time, most of the current tortuosity models are a function of just porosity. Although the model obtained by Yan Han et al. [23] takes into account the effect of particle arrangement on tortuosity, Yun et al. [24] believes that a tortuosity model derived from the hypothesis of square particles is not universal. The calculated value by Yan Han et al. [23] appears larger than estimated by the other tortuosity models in the low porosity section when the particles are arranged at the lower limit (Figure 1), which may be caused by the assumption that the soil particles are square.

**Table 1.** Summary of published tortuosity models.

| Models | Source | Comment |
|---|---|---|
| $\xi = 1 - P \ln \phi$ | reference [15,16] | A function of porosity containing fitting parameters related to particle shape |
| $\xi = \phi^{-\beta}$ | reference [17] | A function of porosity containing fitting parameters |
| $\xi = 1 + 0.8(1 - \phi)$ | reference [18] | A function of porosity containing fitting parameters (0.8) |
| $\xi = 1 + a \frac{1-\phi}{(\phi-\phi_c)^m}$ | reference [19] | A function of porosity containing fitting parameters |
| $\xi = 1.47 e^{-0.3708\phi}$ | reference [20] | A function of porosity containing fitting parameters (1.47) |
| $\xi = \frac{1}{2}\left[1 + \frac{1}{2}\sqrt{1-\phi} + \frac{\sqrt{\left(1-\sqrt{1-\phi}\right)^2 + (1-\phi)/4}}{1-\sqrt{1-\phi}}\right]$ | reference [21] | A function of porosity |
| $\xi = \frac{\phi}{1-(1-\phi)^{2/3}}$ | reference [22] | A function of porosity |

Note: $\xi$ is tortuosity; $\phi$ is porosity; $\phi_c$ is effective porosity; other symbols are fitting parameters.

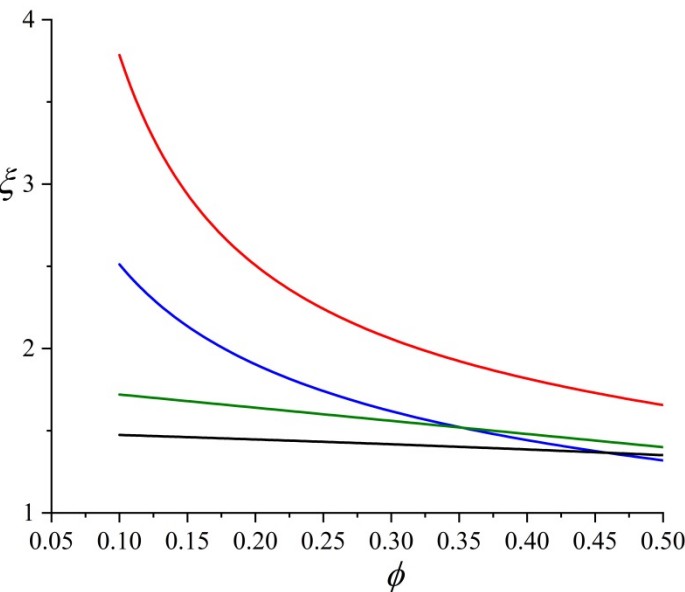

**Figure 1.** Comparison of tortuosity models. (Red curve [23], Blue curve [17], Green curve [18] and Black curve [22]).

In view of the fact that most tortuous models contain fitting parameters only considering the influence of porosity, this paper uses geometric analysis and assumes that the shape of the soil particles is circular to derive a two-dimensional soil geometric tortuosity model with a comprehensive inclusion of porosity and particle arrangement. The model is compared to the results of other published models. Finally, the model and its parameters are analyzed and corresponding conclusions are drawn. The results presented here increase the understanding of the physical mechanisms controlling tortuosity and, hence, the process of fluid seepage through soil, providing a theoretical basis for solving the geotechnical engineering diseases caused by water seepage or air permeability [25,26].

## 2. A Two-Dimensional Geometric Tortuosity Model

The concept of tortuosity is usually defined by the following formula [27]:

$$\xi = \frac{L_t}{L_0} \tag{1}$$

where $\xi$ is tortuosity, $L_t$ is actual streamline length in porous media, and $L_0$ is the length of the straight line corresponding to $L_t$. Thus, tortuosity is a dimensionless number not less than 1.

There are many streamlines in the actual seepage process. Therefore, a soil geometric tortuosity model is obtained by calculating the average tortuosity of representative streamlines around soil particles [21], namely:

$$\xi = \frac{1}{N}\sum_i \xi_i \tag{2}$$

where $N$ is the total number of streamlines, and $\xi_i$ is the tortuosity of the streamline $i$.

It is very difficult to calculate the tortuosity owing to the complexity of soil pore structure. Therefore, the following assumptions are made based on the model established by Yun [24]:

(1) The shape of soil particles is circular with a uniform size;

(2) The passing fluid is Newtonian and in laminar motion.

Soil particles are uniformly distributed and the assumed direction of seepage is from left to right (Figure 2a). $r$ is the particle radius, $C$ is the distance between adjacent particles

parallel to the flow direction, $B$ is the distance between two soil particles perpendicular to the flow direction, and $m = B/C$ is defined as an anisotropic parameter. $\theta$ is the offset angle of two adjacent rows of particles in the vertical flow direction, which is defined as the obstruction parameter ranging from 0 to $\arctan(B/2C)$, where $\theta = 0$ represents the lower limit arrangement (Figure 2b) and $\theta = \arctan(B/2C)$ indicates the upper limit arrangement (Figure 2c). The arrangement of soil particles is determined by the two parameters $m$ and $\theta$ [23].

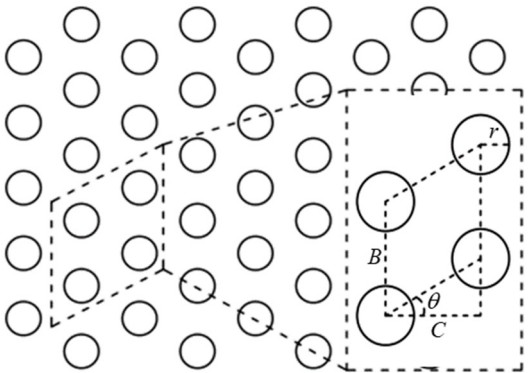

(a) Particle arrangement relation.

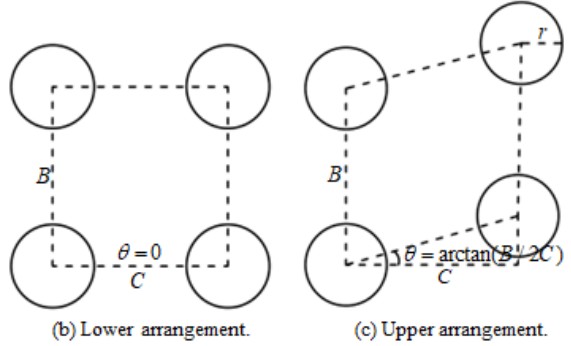

(b) Lower arrangement.　　(c) Upper arrangement.

**Figure 2.** Schematic diagram of particle distribution.

The dotted line in Figure 2b,c indicates the representative unit, and its porosity can be expressed by the following formula:

$$\phi = \frac{A_t - A_s}{A_t} = \frac{BC - \pi r^2}{BC} \tag{3}$$

Therefore:

$$BC = \frac{\pi r^2}{1 - \phi} \tag{4}$$

where $A_t$ is the area of the representative unit, and $A_s$ is the area of the solid part in the representative unit. Combining the above with anisotropic parameters, we obtain:

$$\frac{r}{C} = \sqrt{\frac{(1 - \phi)m}{\pi}} \tag{5}$$

When the fluid flows in from the $B$ side of the unit body with laminar flow state, three different streamlines may appear, as shown in Figure 3.

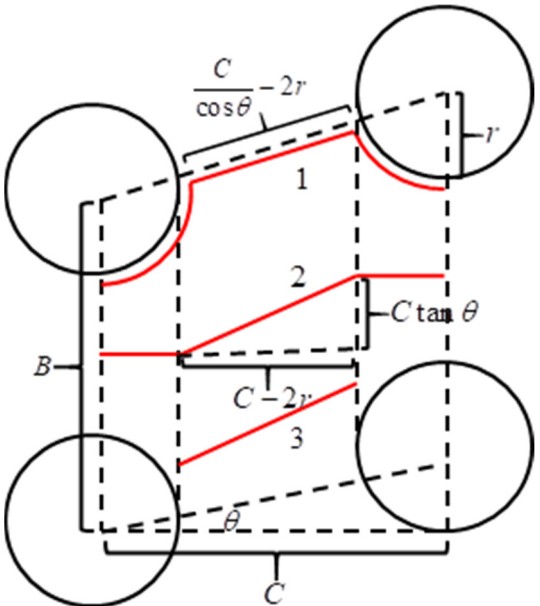

**Figure 3.** Schematic diagram of potential streamlines.

As can be seen in Figure 3, the length of streamline 1 can be represented as:

$$L_1 = \pi r + \frac{C}{\cos \theta} - 2r \tag{6}$$

Combined with Equation (1), $\xi_1$ can be obtained:

$$\xi_1 = \frac{\pi r + \frac{C}{\cos \theta} - 2r}{C} = \frac{(\pi - 2)r \cos \theta + C}{C \cos \theta} \tag{7}$$

The length of streamline 2 can be expressed as

$$L_2 = 2r + \sqrt{(C - 2r)^2 + C^2 \tan^2 \theta} \tag{8}$$

Combined with Equation (1), $\xi_2$ can be obtained:

$$\xi_2 = \frac{2r + \sqrt{(C - 2r)^2 + C^2 \tan^2 \theta}}{C} \tag{9}$$

In reality, the soil particles will overlap each other. When the particles are completely packed, fluid cannot pass through the region on the left and right sides of the unit body, and the actual streamline length is shown by streamline 3 in Figure 2. Combined with Equation (1), $\xi_3$ can be expressed as:

$$\xi_3 = \frac{\sqrt{(C - 2r)^2 + C^2 \tan^2 \theta}}{C - 2r} \tag{10}$$

Combining Equations (2), (6), (8) and (9), a two-dimensional geometric tortuosity model based on soil particle arrangement and porosity can be obtained:

$$\xi = \frac{1}{3}(\xi_1 + \xi_2 + \xi_3) = \frac{1}{3}\left[ \pi \frac{r}{C} + \frac{1}{\cos \theta} + \sqrt{\left(2\frac{r}{C} - 1\right)^2 + \tan^2 \theta} + \frac{\sqrt{\left(2\frac{r}{C} - 1\right)^2 + \tan^2 \theta}}{1 - 2\frac{r}{C}} \right] \tag{11}$$

Putting Equation (4) into Equation (10), it becomes:

$$\xi = \frac{1}{3}\left[\sqrt{\pi m(1-\phi)} + \frac{1}{\cos\theta} + \sqrt{\left(2\sqrt{\frac{m(1-\phi)}{\pi}} - 1\right)^2 + \tan^2\theta} + \frac{\sqrt{\left(2\sqrt{\frac{m(1-\phi)}{\pi}} - 1\right)^2 + \tan^2\theta}}{1 - 2\sqrt{\frac{m(1-\phi)}{\pi}}}\right] \tag{12}$$

Equation (11) is a two-dimensional soil geometric tortuosity model, which includes the comprehensive effects of particles arrangement and porosity. It can be seen that tortuosity is a function determined by porosity and particles arrangement, without the need for empirical constants and that clearly expresses the physical mechanism of tortuosity. The arrangement relationship of particles is described by $m$ and $\theta$ together.

Meanwhile, $1 - 2\sqrt{\frac{m(1-\phi)}{\pi}} > 0$ specifies that the applicable porosity range of the model is $(1 - \pi/4m) \sim 1$, which shows that the applicable range of porosity of the model is related to the parameter $m$. However, the determination of the particle arrangement parameters involves complex microprocesses, so there is no effective method to obtain them. It is found by comparative analysis (see Figures 4–6) that the tortuosity model proposed in this paper is in good agreement with the previous models when $m = 1$. So, bringing $m = 1$ into $(1 - \pi/4m) - 1$ to calculate the applicable porosity range of the model is 0.21–1, which is basically consistent with the value range of soil natural porosity (0.2–1) [28]. According to $\theta = \arctan(B/2C)$ and $m = 1$, the value range of $\theta$ is 0–26.57°.

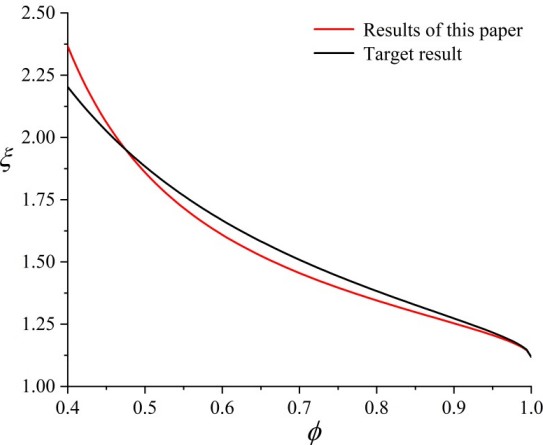

**Figure 4.** Comparison of the proposed model of tortuosity with previously published [23] results—upper limit arrangement.

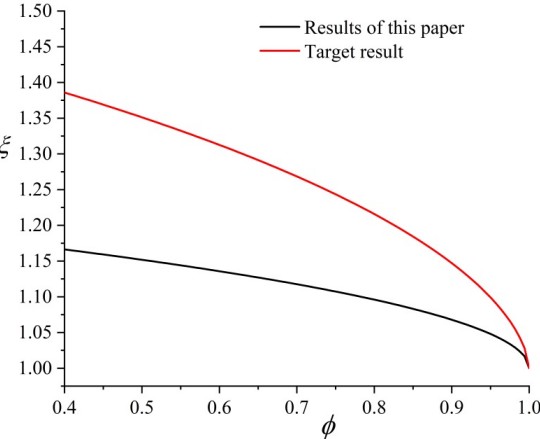

**Figure 5.** Comparison of the proposed model of tortuosity with previously published [22] results—lower limit arrangement.

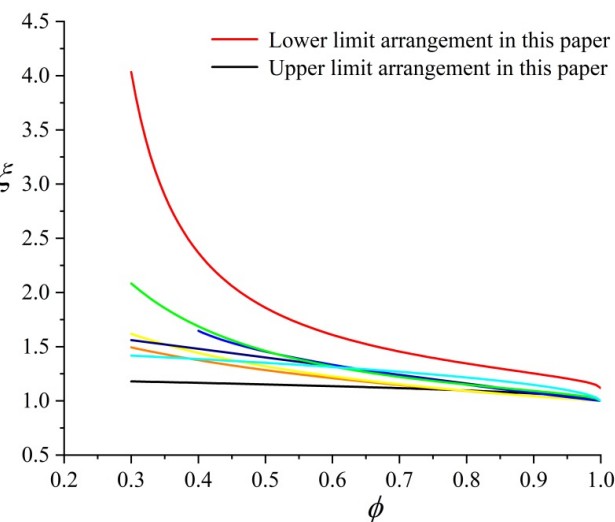

**Figure 6.** Comparison of the tortuosity model in this paper with other models [15–19,21,22].

When the lower limit of soil particles arrangement is applied, $\theta = 0$ in Formula (11), i.e.,

$$\xi = \frac{1}{3}\left[\sqrt{\pi m(1-\phi)} + 1 + \sqrt{\left(2\sqrt{\frac{m(1-\phi)}{\pi}} - 1\right)^2} + \frac{\sqrt{\left(2\sqrt{\frac{m(1-\phi)}{\pi}} - 1\right)^2}}{1 - 2\sqrt{\frac{m(1-\phi)}{\pi}}}\right] \quad (13)$$

When the upper arrangement limit is present, bringing $\theta = \arctan(B/2C)$ into Formula (11) results in:

$$\xi = \frac{1}{3}\left[\sqrt{\pi m(1-\phi)} + \sqrt{1 + \frac{m^2}{4}} + \sqrt{\left(2\sqrt{\frac{m(1-\phi)}{\pi}} - 1\right)^2 + \frac{m^2}{4}} + \frac{\sqrt{\left(2\sqrt{\frac{m(1-\phi)}{\pi}} - 1\right)^2 + \frac{m^2}{4}}}{1 - 2\sqrt{\frac{m(1-\phi)}{\pi}}}\right] \quad (14)$$

Equations (12) and (13) show that when the porosity of soil is constant, the tortuosity is not a fixed value but changes over an interval. The upper limit and lower limit of this interval correspond to the upper limit arrangement and the lower limit arrangement of the soil particles respectively, as shown in Figure 2a,b.

When $\phi \to 0$, the soil particles occupy the whole space, and the particle arrangement can be regarded as the upper limit arrangement. Therefore $\xi \to \infty$ is obtained by Formula (13). When $\phi \to 1$, pores occupy the whole space, so the model becomes closer to the lower limit arrangement, and $\xi \to 1$ is described by Formula (12). These properties of the model are in line with the reality, which preliminarily suggests the reliability of the model.

## 3. Comparison with Published Models

In order to verify the correctness of the model, the upper and lower limit arrangement results of the proposed model were compared with previously published results. The comparisons results are shown in Figures 4 and 5, respectively. The results show that the tortuosity model proposed in this paper has the same changing trends and estimated values (the maximum difference is 0.2, Figure 5) as the previous tortuosity models under the two extreme conditions of soil particle arrangement.

Some classical tortuosity model curves are drawn in Figure 6, and there are obvious differences between them, which is mainly caused by the differences in research methods and research objects. Further analysis of Figure 6 shows that the results from the previous tortuosity models lie between the upper limit arrangement and the lower line arrangement of the model established in this paper and, hence, are special cases of the current model.

## 4. Analysis of Tortuosity Model

As can be seen in Figures 4 and 5, the change of tortuosity with porosity is also different when the arrangement of soil particles is different, which indicates that tortuosity is not only related to porosity but also affected by the arrangement of particles. Figure 6 shows that: (1) The tortuosity shows obvious differences when the porosity is low. This is because when the porosity is small, the particles are in the upper limit arrangement and the flow path will be mostly in the form of streamline 1 (Figure 3), which leads to the enlargement of the flow path, resulting in larger tortuosity, and the curve is concave (Figure 4). When particles are in the lower limit arrangement, there is still a channel for fluid to pass through as the soil particles are round, so the tortuosity is relatively small and the curve is convex (Figure 5); (2) With the increase in porosity, the tortuosity of the upper and lower limit arrangements gradually tends to 1. This is because with the increase in porosity, the internal channels of porous media become larger, and the streamlines will gradually approach the macroscopic straight-line length, which indicates that with the increase in porosity, the influence of particle arrangement on tortuosity gradually decreases.

## 5. Influence of Parameter *m* on Tortuosity

In order to explore the influence of parameter *m* on tortuosity, in the lower limit arrangement (Figure 2), different *m* values are obtained by changing the direction of fluid entering the soil. The operation is as follows: assuming that the medium is isotropic, $m = 1$; if anisotropic, that is, the soil is heterogeneous, when the fluid flows in from *B*, suppose $m = 0.8$, $B = 0.8C$, so when flowing in from *C*, $C = 0.8B$ and $m = 1.25$.

These three different values are taken into Equation (13) respectively, and the results are shown in Figure 7. The tortuosity increases with the increase in *m*. This shows that for the same soil, when the fluid passes through the soil from different sides, the tortuosity is different, resulting in different permeability, which is in line with reality [29,30].

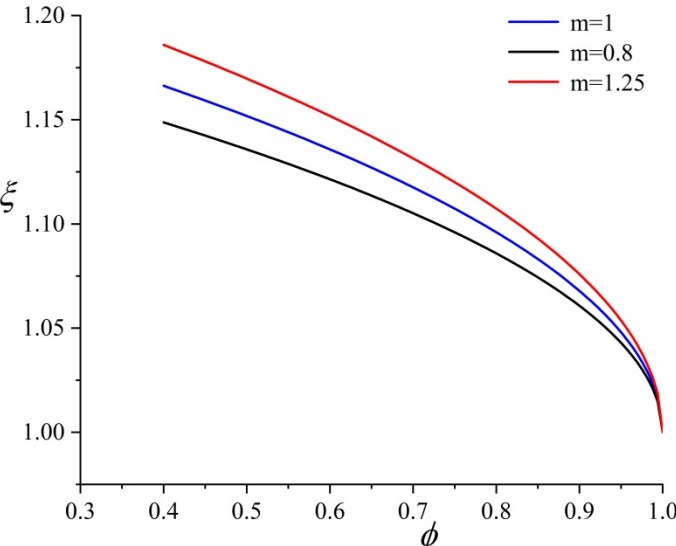

**Figure 7.** Influence of different values of *m* on tortuosity.

## 6. Influence of Parameter $\theta$ on Tortuosity

In Equation (12), the curve of the relationship between tortuosity and porosity with different $\theta$ values can be obtained by assuming $m = 1$ and keeping it unchanged, as shown in Figure 8. The results show that tortuosity is positively correlated with parameter $\theta$.

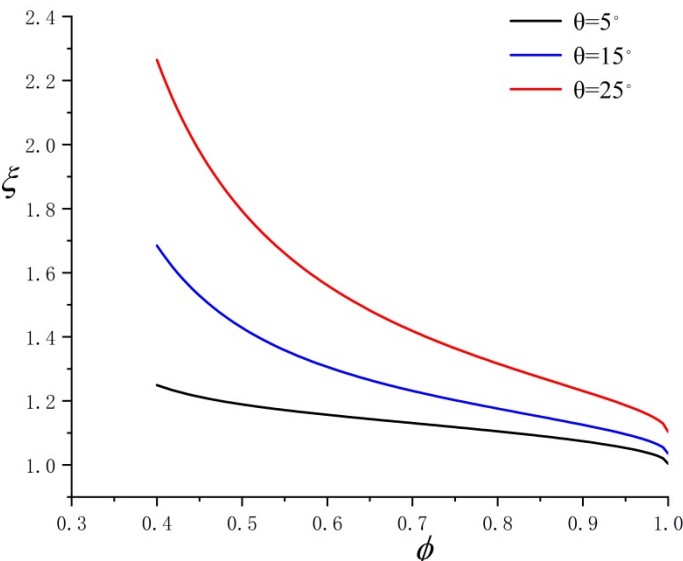

**Figure 8.** Influence of different values of $\theta$ on tortuosity.

As indicated in Figures 7 and 8, the influence of parameters $m$ and $\theta$ at low porosity on tortuosity is significant, then it decreases gradually with the increase in porosity. This confirms that the effect of particle arrangement on tortuosity decreases with the increase in porosity.

## 7. Conclusions

In this paper, a tortuosity model, which included the comprehensive effects of soil particle arrangement and porosity was established by theoretical derivation and geometric analysis. The model compares well with previously published models.

The analysis of the model and its parameters draws the following conclusions:

(1) Besides porosity, soil particle arrangement is another important factor affecting tortuosity. When particles are arranged differently, the tortuosity and porosity of soil can present completely different functional relationships.

(2) With the increase in porosity, the influence of parameters $m$ and $\theta$ on tortuosity gradually decreases, that is, the influence of particle arrangement on tortuosity gradually decreases.

(3) The greater the anisotropic parameter $m$ is, the greater the degree of tortuosity is, which means that for the same soil, when fluid passes through the soil from different sides this will lead to different permeabilities, which is in line with observations from reality; when the porosity is constant, the tortuosity is positively correlated with the parameter $\theta$.

(4) It is suggested that the value of parameter $m$ is 1, so the range of parameter $\theta$ is 0–26.57 °, and the applicable porosity range of the model is 0.21–1, which is consistent with the natural distribution of soil porosity (0.2–1).

The tortuosity model established in this paper only considers the influence of porosity and particle arrangement, and ignores the two main factors of particle shapes and size. Because the establishment of the tortuosity model by geometric method involves the process of quantitative calculation, the calculation results can be obtained for a single standard shape particle, but it is difficult to calculate the results when considering arbitrary irregular shape of particles. At the same time, the size of particles can be characterized by fractal dimension, but a large number of particles of different sizes need to be used in the establishment of geometric model, which makes the calculation process complex, the amount of calculation is too large, and it is difficult to calculate the results. Of course, the establishment of a tortuosity model with the comprehensive effect of multifactors is a more

important research topic, which may be obtained with the improvement of experimental technology and equipment in the future.

**Author Contributions:** Conceptualization: Z.Y., J.G.; methodology: J.G., Z.Y.; formal analysis: Z.Y., J.G., Y.Z.; writing—original draft: J.G., Z.Z., C.L.; writing—review and editing: Y.Y., Y.G., Z.Z., C.L.; resources: R.W., B.Z., Y.F., D.Y., J.Z., H.L., J.S.; funding acquisition: Z.Y., Y.Y.; validation: Z.Y., Y.Y., Y.G., Y.Z. All authors have read and agreed to the published version of the manuscript.

**Funding:** This research was supported by the National Natural Science Foundation of China (Grant No. 41861134008), the Muhammad Asif Khan academician workstation of Yunnan Province (Grant No. 202105AF150076), the Key R&D Program of Yunnan Province (Grant No. 202003AC100002), and the General Program of basic research plan of Yunnan Province (Grant No. 202001AT070043).

**Data Availability Statement:** Data are contained within the article.

**Conflicts of Interest:** The authors declare that they have no known competing financial interests or personal relationships that could have appeared to influence the work reported in this paper.

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
