# Peer review of "Two-Dimensional Soil Geometric Tortuosity Model Based on Porosity and Particle Arrangement"

_minerals, doi:10.3390/min12010043_

Round 1
Reviewer 1 Report
- Several models are mentioned in the introduction section (Table 1 and Figure 1). The authors can enhance the introduction by comparing the performance of these model and highlighting their limitations.
- p. 2, lines 41-46, this section should be strengthened to highlight the research gap the manuscript is addressing and the application of the study
- p.6, line 11, can authors replace the word "overlapped", Soil is closely packed, so overlapped seems inappropriate
- p.7, lines 121 - 132, have the authors conducted an experimental programme to verify the results?
- p.7, lines 140-142, what are the conditions that cause the special cases?
- A strong discussion on the comparison of the results of this paper with published literature is suggested.
- p.8, lines 156-159, how tortuosity will increase when porosity is less? Please explain. Particles can be pushed apart in granular soil but not in fine grained soil. Is the model suitable only for coarse /granular soil?
- p.8, lines 164-166, substantiate with suitable references
- p.9, lines 175-176, how do authors state its in line with reality? Use suitable references or justifications from the model to clarify the statement, explaining in terms of anisotropy.
- Authors should definitely improve the discussion on the results. Now they have only presented the results.
- References from pervious study should be included to augment the results
- Was there an experimental component? Is this not clear in the manuscript. If so, it needs elaboration.
Reviewer 2 Report
Dear editor
I have evaluated the article entitled "Two-dimensional Soil Geometric Tortuosity Model Based on Porosity and Particle Arrangement" submitted to Minerals by Gan et al.
The article propose a 2D geometrical model in order to evaluate the tortuosity of flow path in soil porosity.
I proposed to reject the article based on the following comments:
- as described by Ben Clennell (1997), the concept of tortuosity should be used with great caution. It is not the case in this article, where tortuosity is considered as a fundamental parameter for describing flow in porous media.
- the model is mathematically correct but it is irrelevant because (1) is is only 2D model, (2) the proposed flow paths are geometrically irrealistics and (3) the particle arrangement is very far from the real arrangement of particles in soils, where preferential drains controls the flow.
best regards
Reviewer 3 Report
see attached file

Reviewer 4 Report
The authors propose a two-dimensional soil geometric tortuosity model with a comprehensive inclusion of porosity and particle arrangement. There is a certain degree of innovation, but there are also some problems that need to be resolved.
1) In Chapter 3, it is necessary to analyze the reasons for the difference from the published models.
2) Fig 4-8: These lines should be curves, not broken lines. Because they are drawn by formula, it should be more smooth. Otherwise, whether some information will be omitted.
3) The paper needs to be polished by professionals.
Reviewer 5 Report
The manuscript discusses a geometric tortuosity model including porosity and particle arrangement. The following issues should be addressed:
- The model still assumes that soil particles are circular and uniformly distributed. The authors should comment whether it is possible to introduce parameters characterizing more irregular particle arrangements, such as shape factors, fractal dimension, particle caging, etc. (Adv. Colloid Interface Sci. 2020, 284, 102252; Langmuir 2018, 34(26), 7827-7843; Nanomaterials 2019, 9(7), 921). Some of these parameters have been included in previous models (Chemical Engineering Science 65, 1891-1896, 2010; Fractals 15, 4, 385-390, 2007).
- The reference list should be checked in order to ensure the correctness of the bibliographic details.
Round 2
Reviewer 2 Report
The authors had not answer deeply to the critics i did.
Another time, i advice to do not accept this manuscript for publication.
Author Response
Point: The authors had not answer deeply to the critics i did.
Response: Thank you for your higher requirements for our manuscript. As M. Ben Clennell(1977) said, tortuosity is a term used to describe the sinuosity and interconnectedness of the pore space as it affects transport processes through porous media. It has five meanings: geometrical, electrical, diffusional and hydraulic tortuosity. At the same time, M. Ben Clennell(1977) believes that the concept of the tortuosity can be understood on a micro-scale and a larger scale. The geometric analysis method simplifies the porous media to a simple pore structure, so the tortuosity model established by this method belongs to the concept of larger scale, and it is found to be universally valid when applying to transport in real case (M. Ben Clennell,1977). Of course, as you say, it is more meaningful to establish a 3D tortuosity model which is closer to the actual case, which is also the research trend in the future. Thanks again.
Reviewer 3 Report
I am satisfied with the authors corrections.
Author Response
Thank you very much for your approval.